# Rotting Bandits

**Nir Levine**
Electrical Engineering Department
The Technion
Haifa 32000, Israel
`levin.nir1@gmail.com`

**Koby Crammer**
Electrical Engineering Department
The Technion
Haifa 32000, Israel
`koby@ee.technion.ac.il`

**Shie Mannor**
Electrical Engineering Department
The Technion
Haifa 32000, Israel
`shie@ee.technion.ac.il`

## Abstract

The Multi-Armed Bandits (MAB) framework highlights the trade-off between acquiring new knowledge (Exploration) and leveraging available knowledge (Exploitation). In the classical MAB problem, a decision maker must choose an arm at each time step, upon which she receives a reward. The decision maker's objective is to maximize her cumulative expected reward over the time horizon. The MAB problem has been studied extensively, specifically under the assumption of the arms' rewards distributions being stationary, or quasi-stationary, over time. We consider a variant of the MAB framework, which we termed *Rotting Bandits*, where each arm's expected reward decays as a function of the number of times it has been pulled. We are motivated by many real-world scenarios such as online advertising, content recommendation, crowdsourcing, and more. We present algorithms, accompanied by simulations, and derive theoretical guarantees.

## 1 Introduction

One of the most fundamental trade-offs in stochastic decision theory is the well celebrated Exploration vs. Exploitation dilemma. Should one acquire new knowledge on the expense of possible sacrifice in the immediate reward (Exploration), or leverage past knowledge in order to maximize instantaneous reward (Exploitation)? Solutions that have been demonstrated to perform well are those which succeed in balancing the two. First proposed by Thompson [1933] in the context of drug trials, and later formulated in a more general setting by Robbins [1985], MAB problems serve as a distilled framework for this dilemma. In the classical setting of the MAB, at each time step, the decision maker must choose (pull) between a fixed number of arms. After pulling an arm, she receives a reward which is a realization drawn from the arm's underlying reward distribution. The decision maker's objective is to maximize her cumulative expected reward over the time horizon. An equivalent, more typically studied, is the *regret*, which is defined as the difference between the optimal cumulative expected reward (under full information) and that of the policy deployed by the decision maker.

MAB formulation has been studied extensively, and was leveraged to formulate many real-world problems. Some examples for such modeling are online advertising [Pandey et al., 2007], routing of packets [Awerbuch and Kleinberg, 2004], and online auctions [Kleinberg and Leighton, 2003].

Most past work (Section 6) on the MAB framework has been performed under the assumption that the underlying distributions are stationary, or possibly quasi-stationary. In many real-world scenarios,

this assumption may seem simplistic. Specifically, we are motivated by real-world scenarios where the expected reward of an arm decreases over time instances that it has been pulled. We term this variant *Rotting Bandits*. For motivational purposes, we present the following two examples.

- Consider an online advertising problem where an agent must choose which ad (arm) to present (pull) to a user. It seems reasonable that the effectiveness (reward) of a specific ad on a user would deteriorate over exposures. Similarly, in the content recommendation context, Agarwal et al. [2009] showed that articles' CTR decay over amount of exposures.

- Consider the problem of assigning projects through crowdsourcing systems [Tran-Thanh et al., 2012]. Given that the assignments primarily require human perception, subjects may fall into boredom and their performance would decay (e.g., license plate transcriptions [Du et al., 2013]).

As opposed to the stationary case, where the optimal policy is to always choose some specific arm, in the case of Rotting Bandits the optimal policy consists of choosing different arms. This results in the notion of *adversarial regret* vs. *policy regret* [Arora et al., 2012] (see Section 6). In this work we tackle the harder problem of minimizing the policy regret.

The main contributions of this paper are the following:

- Introducing a novel, real-world oriented MAB formulation, termed *Rotting Bandits*.

- Present an easy-to-follow algorithm for the general case, accompanied with theoretical guarantees.

- Refine the theoretical guarantees for the case of existing prior knowledge on the rotting models, accompanied with suitable algorithms.

The rest of the paper is organized as follows: in Section 2 we present the model and relevant preliminaries. In Section 3 we present our algorithm along with theoretical guarantees for the general case. In Section 4 we do the same for the parameterized case, followed by simulations in Section 5. In Section 6 we review related work, and conclude with a discussion in Section 7.

## 2 Model and Preliminaries

We consider the problem of Rotting Bandits (RB); an agent is given $K$ arms and at each time step $t = 1, 2, ..$ one of the arms must be pulled. We denote the arm that is pulled at time step $t$ as $i(t) \in [K] = \{1, .., K\}$. When arm $i$ is pulled for the $n^{\text{th}}$ time, the agent receives a time independent, $\sigma^2$ sub-Gaussian random reward, $r_t$, with mean $\mu_i(n)$.[1]

In this work we consider two cases: (1) There is no prior knowledge on the expected rewards, except for the 'rotting' assumption to be presented shortly, i.e., a non-parametric case (NPC). (2) There is prior knowledge that the expected rewards comprised of an unknown constant part and a *rotting* part which is known to belong to a set of rotting models, i.e., a parametric case (PC).

Let $N_i(t)$ be the number of pulls of arm $i$ at time $t$ not including this round's choice ($N_i(1) = 0$), and $\Pi$ the set of all sequences $i(1), i(2), ..$, where $i(t) \in [K], \forall t \in \mathbb{N}$. i.e., $\pi \in \Pi$ is an infinite sequence of actions (arms), also referred to as a policy. We denote the arm that is chosen by policy $\pi$ at time $t$ as $\pi(t)$. The objective of an agent is to maximize the expected total reward in time $T$, defined for policy $\pi \in \Pi$ by,

$$J(T; \pi) = \mathbb{E}\left[\sum_{t=1}^{T} \mu_{\pi(t)}\left(N_{\pi(t)}(t) + 1\right)\right] \tag{1}$$

We consider the equivalent objective of minimizing the regret in time $T$ defined by,

$$\mathcal{R}(T; \pi) = \max_{\tilde{\pi} \in \Pi}\{J(T; \tilde{\pi})\} - J(T; \pi). \tag{2}$$

**Assumption 2.1.** *(Rotting)* $\forall i \in [K]$, $\mu_i(n)$ *is positive, and non-increasing in* $n$.

## 2.1 Optimal Policy

Let $\pi^{\mathrm{max}}$ be a policy defined by,

$$\pi^{\mathrm{max}}(t) \in \underset{i \in [K]}{\mathrm{argmax}}\{\mu_i(N_i(t) + 1)\} \tag{3}$$

where, in a case of tie, break it randomly.

**Lemma 2.1.** $\pi^{\mathrm{max}}$ *is an optimal policy for the RB problem.*
***Proof:*** *See Appendix B of the supplementary material.*

## 3 Non-Parametric Case

In the NPC setting for the RB problem, the only information we have is that the expected rewards sequences are positive and non-increasing in the number of pulls. The Sliding-Window Average (**SWA**) approach is a heuristic for ensuring with high probability that, at each time step, the agent did not sample significantly sub-optimal arms too many times. We note that, potentially, the optimal arm changes throughout the trajectory, as Lemma 2.1 suggests. We start by assuming that we know the time horizon, and later account for the case we do not.

### Known Horizon
The idea behind the SWA approach is that after we pulled a significantly sub-optimal arm "enough" times, the empirical average of these "enough" pulls would be distinguishable from the optimal arm for that time step and, as such, given any time step there is a bounded number of significantly sub-optimal pulls compared to the optimal policy. Pseudo algorithm for SWA is given by Algorithm 1.

---
**Algorithm 1** SWA

---
    **Input** : $K, T, \alpha > 0$
    **Initialize** : $M \leftarrow \lceil \alpha 4^{2/3} \sigma^{2/3} K^{-2/3} T^{2/3} \ln^{1/3}(\sqrt{2}T) \rceil$, and $N_i \leftarrow 0$ for all $i \in [K]$
    **for** $t = 1, 2, .., KM$ **do**
        **Ramp up** : $i(t)$ by Round-Robin, receive $r_t$, and set $N_{i(t)} \leftarrow N_{i(t)} + 1$ ; $r_{i(t)}^{N_{i(t)}} \leftarrow r_t$
    **end for**
    **for** $t = KM + 1, ..., T$ **do**
        **Balance** : $i(t) \in \mathrm{argmax}_{i \in [K]} \left\{ \frac{1}{M} \sum_{n=N_i - M + 1}^{N_i} r_i^n \right\}$
        **Update** : receive $r_t$, and set $N_{i(t)} \leftarrow N_{i(t)} + 1$ ; $r_{i(t)}^{N_{i(t)}} \leftarrow r_t$
    **end for**

---

**Theorem 3.1.** *Suppose Assumption 2.1 holds. SWA algorithm achieves regret bounded by,*

$$\mathcal{R}(T; \pi^{\mathrm{SWA}}) \leq \left( \alpha \max_{i \in [K]} \mu_i(1) + \alpha^{-1/2} \right) 4^{2/3} \sigma^{2/3} K^{1/3} T^{2/3} \ln^{1/3}(\sqrt{2}T) + 3K \max_{i \in [K]} \mu_i(1)$$
$$\tag{4}$$

***Proof:*** *See Appendix C.1 of the supplementary material.*

We note that the upper bound obtains its minimum for $\alpha = \left(2 \max_{i \in [K]} \mu_i(1)\right)^{-2/3}$, which can serve as a way to choose $\alpha$ if $\max_{i \in [K]} \mu_i(1)$ is known, but $\alpha$ can also be given as an input to SWA to allow control on the averaging window size.

### Unknown Horizon
In this case we use *doubling trick* in order to achieve the same horizon-dependent rate for the regret. We apply the SWA algorithm with a series of increasing horizons (powers of two, i.e., $1, 2, 4, ..$) until reaching the (unknown) horizon. We term this Algorithm wSWA (wrapper SWA).

**Corollary 3.1.1.** *Suppose Assumption 2.1 holds. wSWA algorithm achieves regret bounded by,*

$$\mathcal{R}(T; \pi^{\mathrm{wSWA}}) \leq \left( \alpha \max_{i \in [K]} \mu_i(1) + \alpha^{-1/2} \right) 8 \sigma^{2/3} K^{1/3} T^{2/3} \ln^{1/3}(\sqrt{2}T)$$
$$+ 3K \max_{i \in [K]} \mu_i(1)(\log_2 T + 1) \tag{5}$$

***Proof:*** *See Appendix C.2 of the supplementary material.*

## 4 Parametric Case

In the PC setting for the RB problem, there is prior knowledge that the expected rewards comprised of a sum of an unknown constant part and a rotting part known to belong to a set of models, $\Theta$. i.e., the expected reward of arm $i$ at its $n^{\text{th}}$ pull is given by, $\mu_i(n) = \mu_i^c + \mu(n; \theta_i^*)$, where $\theta_i^* \in \Theta$. We denote $\{\theta_i^*\}_{i=1}^{[K]}$ by $\Theta^*$. We consider two cases: The first is the asymptotically vanishing case (AV), i.e., $\forall i : \mu_i^c = 0$. The second is the asymptotically non-vanishing case (ANV), i.e., $\forall i : \mu_i^c \in \mathbb{R}$.

We present a few definitions that will serve us in the following section.

**Definition 4.1.** *For a function $f : \mathbb{N} \to \mathbb{R}$, we define the function $f^{\star\downarrow} : \mathbb{R} \to \mathbb{N} \cup \{\infty\}$ by the following rule: given $\zeta \in \mathbb{R}$, $f^{\star\downarrow}(\zeta)$ returns the smallest $N \in \mathbb{N}$ such that $\forall n \geq N : f(n) \leq \zeta$, or $\infty$ if such $N$ does not exist.*

**Definition 4.2.** *For any $\theta_1 \neq \theta_2 \in \Theta^2$, define $det_{\theta_1,\theta_2}, Ddet_{\theta_1,\theta_2} : \mathbb{N} \to \mathbb{R}$ as,*

$$det_{\theta_1,\theta_2}(n) = \frac{n\sigma^2}{\left(\sum_{j=1}^n \mu(j;\theta_1) - \sum_{j=1}^n \mu(j;\theta_2)\right)^2}$$

$$Ddet_{\theta_1,\theta_2}(n) = \frac{n\sigma^2}{\left(\sum_{j=1}^{\lfloor n/2 \rfloor} [\mu(j;\theta_1) - \mu(j;\theta_2)] - \sum_{j=\lfloor n/2 \rfloor+1}^n [\mu(j;\theta_1) - \mu(j;\theta_2)]\right)^2}$$

**Definition 4.3.** *Let $bal : \mathbb{N} \cup \infty \to \mathbb{N} \cup \infty$ be defined at each point $n \in \mathbb{N}$ as the solution for,*

$$\min \ \alpha \qquad \text{s.t, } \max_{\theta \in \Theta} \mu(\alpha; \theta) \leq \min_{\theta \in \Theta} \mu(n; \theta)$$

*We define $bal(\infty) = \infty$.*

**Assumption 4.1.** *(**Rotting Models**) $\mu(n;\theta)$ is positive, non-increasing in $n$, and $\mu(n;\theta) \in o(1)$, $\forall \theta \in \Theta$, where $\Theta$ is a discrete known set.*

We present an example for which, in Appendix E, we demonstrate how the different following assumptions hold. By this we intend to achieve two things: (i) show that the assumptions are not too harsh, keeping the problem relevant and non-trivial, and (ii) present a simple example on how to verify the assumptions.

**Example 4.1.** *The reward of arm $i$ for its $n^{th}$ pull is distributed as $\mathcal{N}\left(\mu_i^c + n^{-\theta_i^*}, \sigma^2\right)$. Where $\theta_i^* \in \Theta = \{\theta_1, \theta_2, ..., \theta_M\}$, and $\forall \theta \in \Theta : 0.01 \leq \theta \leq 0.49$.*

### 4.1 Closest To Origin (AV)

The Closest To Origin (CTO) approach for RB is a heuristic that simply states that we hypothesize that the true underlying model for an arm is the one that best fits the past rewards. The fitting criterion is proximity to the origin of the sum of expected rewards shifted by the observed rewards. Let $r_1^i, r_2^i, .., r_{N_i(t)}^i$ be the sequence of rewards observed from arm $i$ up until time $t$. Define,

$$Y(i, t; \Theta) = \left\{ \sum_{j=1}^{N_i(t)} r_j^i - \sum_{j=1}^{N_i(t)} \mu(j; \theta) \right\}_{\theta \in \Theta}. \tag{6}$$

The CTO approach dictates that at each decision point, we assume that the true underlying rotting model corresponds to the following proximity to origin rule (hence the name),

$$\hat{\theta}_i(t) = \underset{\theta \in \Theta}{\operatorname{argmin}} \{|Y(i, t; \theta)|\}. \tag{7}$$

The CTO$_{\text{SIM}}$ version tackles the RB problem by simultaneously detecting the true rotting models and balancing between the expected rewards (following Lemma 2.1). In this approach, every time step, each arm's rotting model is hypothesized according to the proximity rule (7). Then the algorithm simply follows an argmax rule, where least number of pulls is used for tie breaking (randomly between an equal number of pulls). Pseudo algorithm for CTO$_{\text{SIM}}$ is given by Algorithm 2.

**Assumption 4.2.** *(**Simultaneous Balance and Detection ability**)*

$$bal\left(\max_{\theta_1 \neq \theta_2 \in \Theta^2}\left\{det_{\theta_1,\theta_2}^{\star\downarrow}\left(\frac{1}{16}\ln^{-1}(\zeta)\right)\right\}\right) \in o(\zeta)$$

The above assumption ensures that, starting from some horizon $T$, the underlying models could be distinguished from the others, w.p $1 - 1/T^2$, by their sums of expected rewards, and the arms could then be balanced, all within the horizon.

**Theorem 4.1.** *Suppose Assumptions 4.1 and 4.2 hold. There exists a finite step $T^*_{\text{SIM}}$, such that for all $T \geq T^*_{\text{SIM}}$, CTO$_{\text{SIM}}$ achieves regret upper bounded by $\mathbf{o}\left(1\right)$ (which is upper bounded by $\max_{\theta \in \Theta^*} \mu\left(1; \theta\right)$). Furthermore, $T^*_{\text{SIM}}$ is upper bounded by the solution for the following,*

$$\min T$$

$$s.t \begin{cases} T, b \in \mathbb{N} \cup \{0\}, t \in \mathbb{N}^K \\ \forall b, \exists t : \begin{cases} \|t\|_1 \leq T + b \\ t_i \geq \max_{\theta \in \Theta^*} \left\{ m^* \left( \frac{1}{K(T+b)^2}; \theta \right) \right\} \\ \mu\left(t_i + 1; \theta_i^*\right) \leq \min_{\tilde{\theta} \in \Theta} \left[ \mu \left( \max_{\theta \in \Theta^*} \left\{ m^* \left( \frac{1}{K(T+b)^2}; \theta \right) \right\}; \tilde{\theta} \right) \right] \end{cases} \end{cases} \quad (8)$$

***Proof****: See Appendix D.1 of the supplementary material.*

Regret upper bounded by $o\left(1\right)$ is achieved by proving that w.p of $1 - 1/T$ the regret vanishes, and in any case it is still bounded by a decaying term. The shown optimization bound stems from ensuring that the arms would be pulled enough times to be correctly detected, and then balanced (following the optimal policy, Lemma 2.1). Another upper bound for $T^*_{\text{SIM}}$ can be found in Appendix D.1.

### 4.2 Differences Closest To Origin (ANV)

We tackle this problem by estimating both the rotting models and the constant terms of the arms. The Differences Closest To Origin (**D-CTO**) approach is composed of two stages: first, detecting the underlying rotting models, then estimating and controlling the pulls due to the constant terms. We denote $a^* = \text{argmax}_{i \in [K]} \{\mu_i^c\}$, and $\Delta_i = \mu_{a^*}^c - \mu_i^c$.

**Assumption 4.3.** *(D-Detection ability)*

$$\max_{\theta_1 \neq \theta_2 \in \Theta^2} \left\{ Ddet^{\star\downarrow}_{\theta_1, \theta_2}\left(\epsilon\right) \right\} \leq D\left(\epsilon\right) < \infty, \quad \forall \epsilon > 0$$

This assumption ensures that for any given probability, the models could be distinguished, by the differences (in pulls) between the first and second halves of the models' sums of expected rewards.

**Models Detection**
In order to detect the underlying rotting models, we cancel the influence of the constant terms. Once we do this, we can detect the underlying models. Specifically, we define a criterion of proximity to the origin based on differences between the halves of the rewards sequences, as follows: define,

$$Z\left(i, t; \Theta\right) = \left( \sum_{j=1}^{\lfloor N_i(t)/2 \rfloor} r_j^i - \sum_{j=\lfloor N_i(t)/2 \rfloor+1}^{N_i(t)} r_j^i \right) - \left( \sum_{j=1}^{\lfloor N_i(t)/2 \rfloor} \mu\left(j; \theta\right) - \sum_{j=\lfloor N_i(t)/2 \rfloor+1}^{N_i(t)} \mu\left(j; \theta\right) \right).$$
$$(9)$$

The D-CTO approach is that in each decision point, we assume that the true underlying model corresponds to the following rule,

$$\hat{\theta}_i\left(t\right) = \underset{\theta \in \Theta}{\text{argmin}} \{|Z\left(i, t; \theta\right)|\} \quad (10)$$

We define the following optimization problem, indicating the number of samples required for ensuring correct detection of the rotting models w.h.p. For some arm $i$ with (unknown) rotting model $\theta_i^*$,

$$\min m \quad \text{s.t} \begin{cases} P\left( \hat{\theta}_i\left(l\right) \neq \theta_i^* \right) \leq p, \quad \forall l \geq m \\ \text{while pulling only arm } i. \end{cases} \quad (11)$$

We denote the solution to the above problem, when we use proximity rule (10), by $m^*_{\text{diff}}\left(p; \theta_i^*\right)$, and define $m^*_{\text{diff}}\left(p\right) = \max_{\theta \in \Theta} \{m^*_{\text{diff}}\left(p; \theta\right)\}$.

| **Algorithm 2** CTO$_{\text{SIM}}$ | **Algorithm 3** D-CTO$_{\text{UCB}}$ |
|---|---|
| **Input** : $K, \Theta$<br>**Initialization** : $N_i = 0, \ \forall i \in [K]$<br>**for** $t = 1, 2, .., K$ **do**<br>   **Ramp up** : $i(t) = t$ ,and update $N_{i(t)}$<br>**end for**<br>**for** $t = K + 1, ...,$ **do**<br>   **Detect** : determine $\{\hat\theta_i\}$ by Eq. (7)<br>   **Balance** : $i(t) \in \text{argmax}_{i \in [K]} \, \mu\left(N_i + 1; \hat\theta_i\right)$<br>   **Update** : $N_{i(t)} \leftarrow N_{i(t)} + 1$<br>**end for** | **Input** : $K, \Theta, \delta$<br>**Initialization** : $N_i = 0, \ \forall i \in [K]$<br>**for** $t = 1, 2, .., K \times m_{\text{diff}}^*(\delta/K)$ **do**<br>   **Explore** :<br>   $i(t)$ by Round Robin, update $N_{i(t)}$<br>**end for**<br>**Detect** : determine $\{\hat\theta_i\}$ by Eq. (10)<br>**for** $t = K \times m_{\text{diff}}^*(\delta/K) + 1, ...,$ **do**<br>   **UCB** : $i(t)$ according to Eq. (12)<br>   **Update** : $N_{i(t)} \leftarrow N_{i(t)} + 1$<br>**end for** |

## D-CTO$_{\text{UCB}}$

We next describe an approach with one decision point, and later on remark on the possibility of having a decision point at each time step. As explained above, after detecting the rotting models, we move to tackle the constant terms aspect of the expected rewards. This is done in a UCB1-like approach [Auer et al., 2002a]. Given a sequence of rewards from arm $i$, $\{r_k^i\}_{k=1}^{N_i(t)}$, we modify them using the estimated rotting model $\hat\theta_i$, then estimate the arm's constant term, and finally choose the arm with the highest estimated expected reward, plus an upper confident term. i.e., at time $t$, we pull arm $i(t)$, according to the rule,

$$i(t) \in \underset{i \in [K]}{\text{argmax}} \left[ \hat\mu_i^c(t) + \mu\left(N_i(t) + 1; \hat\theta_i(t)\right) + c_{t, N_i(t)} \right] \tag{12}$$

where $\hat\theta_i(t)$ is the estimated rotting model (obtained in the first stage), and,

$$\hat\mu_i^c(t) = \frac{\sum_{j=1}^{N_i(t)} \left( r_j^i - \mu\left(j; \hat\theta_i(t)\right) \right)}{N_i(t)}, \qquad c_{t,s} = \sqrt{\frac{8 \ln(t)\, \sigma^2}{s}}$$

In a case of a tie in the UCB step, it may be arbitrarily broken. Pseudo algorithm for D-CTO$_{\text{UCB}}$ is given by Algorithm 3, accompanied with the following theorem.

**Theorem 4.2.** *Suppose Assumptions 4.1, and 4.3 hold. For $\delta \in (0,1)$, with probability of at least $1 - \delta$,* D-CTO$_{\text{UCB}}$ *algorithm achieves regret bounded at time $T$ by,*

$$\sum_{\substack{i \in [K] \\ i \neq a^*}} \left[ \max\left\{ m_{\text{diff}}^*(\delta/K), \mu^{\star\downarrow}(\epsilon_i; \theta_i^*), \frac{32\sigma^2 \ln T}{(\Delta_i - \epsilon_i)^2} \right\} \times (\Delta_i + \mu(1; \theta_{a^*}^*)) \right] + C(\Theta^*, \{\mu_i^c\}) \tag{13}$$

*for any sequence $\epsilon_i \in (0, \Delta_i), \forall i \neq a^*$. Where $\frac{32\sigma^2 \ln T}{(\Delta_i - \epsilon_i)^2}$ is the only time-dependent factor.*
**Proof:** *See Appendix D.2 of the supplementary material.*

A few notes on the result: Instead of calculating $m_{\text{diff}}^*(\delta/K)$, it is possible to use any upper bound (e.g., as shown in Appendix E, $\max_{\theta_1 \neq \theta_2 \in \Theta^2} Ddet_{\theta_1, \theta_2}^{\star\downarrow}\left(\frac{1}{8}\ln^{-1}\left(\frac{2K}{\delta}\right)\right)$ rounded to higher even number). We cannot hope for a better rate than $\ln T$ as stochastic MAB is a special case of the RB problem. Finally, we can convert the D-CTO$_{\text{UCB}}$ algorithm to have a decision point in each step: at each time step, determine the rotting models according to proximity rule (10), followed by pulling an arm according to Eq. (12). We term this version D-CTO$_{\text{SIM-UCB}}$.

## 5 Simulations

We next compare the performance of the SWA and CTO approaches with benchmark algorithms.

**Setups** for all the simulations we use Normal distributions with $\sigma^2 = 0.2$, and $T = 30,000$.
*Non-Parametric:* $K = 2$. As for the expected rewards: $\mu_1(n) = 0.5, \forall n$, and $\mu_2(n) = 1$ for its first $7,500$ pulls and $0.4$ afterwards. This setup is aimed to show the importance of not relying on the

Table 1: Number of 'wins' and p-values between the different algorithms

| | | UCB1 | DUCB | SWUCB | wSWA | (D-)CTO |
|---|---|---|---|---|---|---|
| **NP** | UCB1 | | <1e-5 | <1e-5 | <1e-5 | |
| | DUCB | 100 | | <1e-5 | <1e-5 | |
| | SWUCB | 100 | 100 | | <1e-5 | |
| | wSWA | **100** | **100** | **100** | | |
| **AV** | UCB1 | | 0.81 | <1e-5 | <1e-5 | <1e-5 |
| | DUCB | 55 | | <1e-5 | <1e-5 | <1e-5 |
| | SWUCB | 15 | 22 | | <1e-5 | <1e-5 |
| | wSWA | **98** | **99** | **100** | | <1e-5 |
| | CTO | **100** | **100** | **100** | **100** | |
| **ANV** | UCB1 | | 0.54 | 0.83 | <1e-5 | <1e-5 |
| | DUCB | 40 | | 0.91 | < 1e-5 | <1e-5 |
| | SWUCB | 50 | 50 | | <1e-5 | <1e-5 |
| | wSWA | **97** | **98** | **97** | | <1e-5 |
| | D-CTO | **100** | **100** | **100** | **66** | |

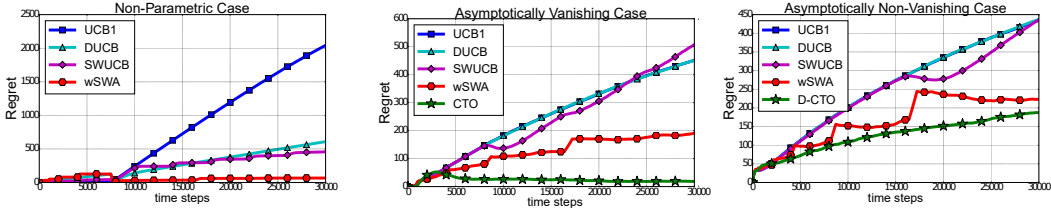

Figure 1: Average regret. Left: non-parametric. Middle: parametric AV. Right: parametric ANV

whole past rewards in the RB setting.

*Parametric AV & ANV:* $K = 10$. The rotting models are of the form $\mu\left(j; \theta\right) = \left(\mathbf{int}\left(\frac{j}{100}\right) + 1\right)^{-\theta}$, where $\mathbf{int}(\cdot)$ is the lower rounded integer, and $\Theta = \{0.1, 0.15, .., 0.4\}$ (i.e., plateaus of length 100, with decay between plateaus according to $\theta$). $\{\theta_i^*\}_{i=1}^K$ were sampled with replacement from $\Theta$, independently across arms and trajectories. $\{\mu_i^c\}_{i=1}^K$ (ANV) were sampled randomly from $[0, 0.5]^K$.

**Algorithms** we implemented standard benchmark algorithms for non-stationary MAB: UCB1 by Auer et al. [2002a], Discounted UCB (DUCB) and Sliding-Window UCB (SWUCB) by Garivier and Moulines [2008]. We implemented CTO$_{SIM}$, D-CTO$_{SIM-UCB}$, and wSWA for the relevant setups. We note that adversarial benchmark algorithms are not relevant in this case, as the rewards are unbounded.

**Grid Searches** were performed to determine the algorithms' parameters. For DUCB, following Kocsis and Szepesvári [2006], the discount factor was chosen from $\gamma \in \{0.9, 0.99, .., 0.999999\}$, the window size for SWUCB from $\tau \in \{1e3, 2e3, .., 20e3\}$, and $\alpha$ for wSWA from $\{0.2, 0.4, .., 1\}$.

**Performance** for each of the cases, we present a plot of the average regret over 100 trajectories, specify the number of 'wins' of each algorithm over the others, and report the p-value of a paired T-test between the (end of trajectories) regrets of each pair of algorithms. For each trajectory and two algorithms, the *'winner'* is defined as the algorithm with the lesser regret at the end of the horizon.

**Results** the parameters that were chosen by the grid search are as follows: $\gamma = 0.999$ for the non-parametric case, and $0.999999$ for the parametric cases. $\tau = 4e3, 8e3$, and $16e3$ for the non-parametric, AV, and ANV cases, respectively. $\alpha = 0.2$ was chosen for all cases.
The average regret for the different algorithms is given by Figure 1. Table 1 shows the number of 'wins' and p-values. The table is to be read as the following: the entries under the diagonal are the number of times the algorithms from the left column 'won' against the algorithms from the top row, and the entries above the diagonal are the p-values between the two.
While there is no clear 'winner' between the three benchmark algorithms across the different cases, wSWA, which does not require any prior knowledge, consistently and significantly outperformed them. In addition, when prior knowledge was available and CTO$_{SIM}$ or D-CTO$_{UCB-SIM}$ could be deployed, they outperformed all the others, including wSWA.

# 6   Related Work

We turn to reviewing related work while emphasizing the differences from our problem.

**Stochastic MAB** In the stochastic MAB setting [Lai and Robbins, 1985], the underlying reward distributions are stationary over time. The notion of regret is the same as in our work, but the optimal policy in this setting is one that pulls a fixed arm throughout the trajectory. The two most common approaches for this problem are: constructing Upper Confidence Bounds which stem from the seminal work by Gittins [1979] in which he proved that index policies that compute upper confidence bounds on the expected rewards of the arms are optimal in this case (e.g., see Auer et al. [2002a], Garivier and Cappé [2011], Maillard et al. [2011]), and Bayesian heuristics such as Thompson Sampling which was first presented by Thompson [1933] in the context of drug treatments (e.g., see Kaufmann et al. [2012], Agrawal and Goyal [2013], Gopalan et al. [2014]).

**Adversarial MAB** In the Adversarial MAB setting (also referred to as the Experts Problem, see the book of Cesa-Bianchi and Lugosi [2006] for a review), the sequence of rewards are selected by an adversary (i.e., can be arbitrary). In this setting the notion of *adversarial regret* is adopted [Auer et al., 2002b, Hazan and Kale, 2011], where the regret is measured against the best possible fixed action that could have been taken in hindsight. This is as opposed to the *policy regret* we adopt, where the regret is measured against the best sequence of actions in hindsight.

**Hybrid models** Some past work consider settings between the Stochastic and the Adversarial settings. Garivier and Moulines [2008] consider the case where the reward distributions remain constant over epochs and change arbitrarily at unknown time instants, similarly to Yu and Mannor [2009] who consider the same setting, only with the availability of side observations. Chakrabarti et al. [2009] consider the case where arms can expire and be replaced with new arms with arbitrary expected reward, but as long as an arm does not expire its statistics remain the same.

**Non-Stationary MAB** Most related to our problem is the so-called Non-Stationary MAB. Originally proposed by Jones and Gittins [1972], who considered a case where the reward distribution of a chosen arm can change, and gave rise to a sequence of works (e.g., Whittle et al. [1981], Tekin and Liu [2012]) which were termed *Restless Bandits* and *Rested Bandits*. In the *Restless Bandits* setting, termed by Whittle [1988], the reward distributions change in each step according to a known stochastic process. Komiyama and Qin [2014] consider the case where each arm decays according to a linear combination of decaying basis functions. This is similar to our parametric case in that the reward distributions decay according to possible models, but differs fundamentally in that it belongs to the *Restless Bandits* setup (ours to the *Rested Bandits*). More examples in this line of work are Slivkins and Upfal [2008] who consider evolution of rewards according to Brownian motion, and Besbes et al. [2014] who consider bounded total variation of expected rewards. The latter is related to our setting by considering the case where the total variation is bounded by a constant, but significantly differs by that it considers the case where the (unknown) expected rewards sequences are not affected by actions taken, and in addition requires bounded support as it uses the EXP3 as a sub-routine. In the *Rested Bandits* setting, only the reward distribution of a chosen arm changes, which is the case we consider. An optimal control policy (reward processes are known, no learning required) to bandits with non-increasing rewards and discount factor was previously presented (e.g., Mandelbaum [1987], and Kaspi and Mandelbaum [1998]). Heidari et al. (2016) consider the case where the reward decays (as we do), but with no statistical noise (deterministic rewards), which significantly simplifies the problem. Another somewhat closely related setting is suggested by Bouneffouf and Feraud [2016], in which statistical noise exists, but the expected reward shape is known up to a multiplicative factor.

# 7   Discussion

We introduced a novel variant of the *Rested Bandits* framework, which we termed *Rotting Bandits*. This setting deals with the case where the expected rewards generated by an arm decay (or generally do not increase) as a function of pulls of that arm. This is motivated by many real-world scenarios.

We first tackled the non-parametric case, where there is no prior knowledge on the nature of the decay. We introduced an easy-to-follow algorithm accompanied by theoretical guarantees.

We then tackled the parametric case, and differentiated between two scenarios: expected rewards decay to zero (AV), and decay to different constants (ANV). For both scenarios we introduced

suitable algorithms with stronger guarantees than for the non-parametric case: For the AV scenario we introduced an algorithm for ensuring, in expectation, regret upper bounded by a term that decays to zero with the horizon. For the ANV scenario we introduced an algorithm for ensuring, with high probability, regret upper bounded by a horizon-dependent rate which is optimal for the stationary case.

We concluded with simulations that demonstrated our algorithms' superiority over benchmark algorithms for non-stationary MAB. We note that since the RB setting is novel, there are not suitable available benchmarks, and so this paper also serves as a benchmark.

For future work we see two main interesting directions: (i) show a lower bound on the regret for the non-parametric case, and (ii) extend the scope of the parametric case to continuous parameterization.

**Acknowledgment**   The research leading to these results has received funding from the European Research Council under the European Union's Seventh Framework Program (FP/2007-2013) / ERC Grant Agreement n. 306638

## Footnotes

[1]Our results hold for pulls-number dependent variances $\sigma^2(n)$, by upper bound them $\sigma^2 \geq \sigma^2(n), \forall n$. It is fairly straightforward to adapt the results to pulls-number dependent variances, but we believe that the way presented conveys the setting in the clearest way.

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
