[Supplementary Material · rotting_bandits_supp.pdf]

## A  Hoeffding's Inequality for Sub-Gaussian RVs

Let $X_1, .., X_n$ be independent, mean-zero, $\sigma_i^2$-sub-Gaussian random variables. Then for all $t \geq 0$,

$$\mathbb{P}\left(\sum_{i=1}^{n} X_i \geq t\right) \leq \exp\left\{-\frac{t^2}{2\sum_{i=1}^{n}\sigma_i^2}\right\} \tag{14}$$

## B  Optimal Policy

### B.1  Proof of Lemma 2.1

In this section we show that $\pi^{\max}$, defined by Eq. (3) is an optimal policy for the RB problem. Assume on the contrary, that $\pi^{\max}$ is not an optimal policy. Thus, there exists a time horizon, $T$, for which there exists some other policy $\pi^{\text{cand}}$ that satisfies $J\left(T; \pi^{\text{cand}}\right) > J\left(T; \pi^{\max}\right)$.

Let $m$ be the first time step in which $\pi^{\text{cand}}$ deviates from $\pi^{\max}$, since $J\left(T; \pi^{\text{cand}}\right) > J\left(T; \pi^{\max}\right)$ we infer that $m \leq T$ (i.e., there is such time step). Let $\tilde{\pi}$ be a policy defined by,

$$\tilde{\pi}(t) = \begin{cases} \pi^{\text{cand}}(t), & \text{if } t < m \\ \operatorname{argmax}_{i \in [K]}\{\mu\left(N_i(m) + 1; \theta_i^*\right)\}, & \text{if } t = m \\ \pi^{\text{cand}}(t-1), & \text{if } t > m \end{cases}$$

where if there exist more than one member in $\operatorname{argmax}_{i \in [K]}\{\mu\left(N_i(m) + 1; \theta_i^*\right)\}$, $\tilde{\pi}$ chooses the same action as $\pi^{\max}$. That is, $\tilde{\pi}$ mimics $\pi^{\text{cand}}$ until time step $m$, then plays according to $\operatorname{argmax}$ rule, and then re-mimics $\pi^{\text{cand}}$. Let $\mu_m, \mu_T$ be the expected rewards of the arms that $\tilde{\pi}$ chose at the $m^{th}$ time step, and that $\pi^{\text{cand}}$ chose at the $T^{th}$ time step, respectively. It is easy to see that,

$$J\left(T; \tilde{\pi}\right) - J\left(T; \pi^{cand}\right) = \mu_m - \mu_T \geq 0 \tag{15}$$

where the second transition holds by the $\operatorname{argmax}$ rule combined with the assumption that the expected rewards are non-increasing (assumption 2.1). Thus, $J\left(T; \tilde{\pi}\right) \geq J\left(T; \pi^{\text{cand}}\right)$. If we apply the above logic steps recursively, we obtain a series of policies with non-decreasing values of expected total reward $J\left(T; \cdot\right)$, where the series ends when there is no time step which deviates from $\pi^{\max}$, i.e., $J\left(T; \pi^{\max}\right) \geq J\left(T; \pi^{\text{cand}}\right)$, in contradiction to $\pi^{\max}$ being non-optimal. Thus, we infer that $\pi^{\max}$ is indeed an optimal policy.

## C  Non-Parametric Case

### C.1  Proof of Thm. 3.1

We define,

$$\begin{cases} M &= \lceil \alpha 4^{2/3}\sigma^{2/3}K^{-2/3}T^{2/3}\ln^{1/3}\left(\sqrt{2}T\right)\rceil \\ q &= \alpha^{-1/2}2^{1/3}\sigma^{2/3}K^{1/3}T^{-1/3}\ln^{1/3}\left(\sqrt{2}T\right) \end{cases}$$

and start by making two useful observations:

Observation 1: By Hoeffding's Inequality we have,

$$P\left(|\bar{X}_M - \mathbb{E}\left[\bar{X}_M\right]| \geq q\right) \leq \frac{1}{T^2} \tag{16}$$

where $\bar{X}_M$ is the empirical average of $M$ independent $\sigma^2$ sub-Gaussian samples.

Observation 2: Since the expected rewards of an arm only depends on the time it is being pulled (and not on the time step itself), the expected total reward of a policy only depends on the number of pulls of the different arms (and not on the order of pulls).

From now on we assume that $|\bar{X}_M - \mathbb{E}\left[\bar{X}_M\right]| < q$ (see Observation 1) for all arms throughout the trajectory, and later address the case where it is violated.

**Step 1:** bound the number of significantly sub-optimal pulls.

In what is following we prove by induction that for all the ends of time steps $t \in [T]$, by applying SWA, there is no arm $j$ for which,

$$\left\{ |n| : \mu_j \left( N_j^{\pi^{\text{SWA}}}(t) - n \right) < \max_{i \in [K]} \left[ \mu_i \left( N_i^{\pi^{\text{SWA}}}(t) \right) \right] - 2q, \quad n \in \mathbb{N}_0 \right\} > M \qquad (17)$$

where $N_i^{\pi^{\text{SWA}}}(t)$ is the number of pulls of arms $i$ at time $t$ induced by policy $\pi^{\text{SWA}}$, which is defined by the SWA algorithm. That is, following SWA ensures that for all time steps, no arm would be pulled more than $M$ times in which its expected reward is at least $2q$ lower than the expected reward of the (current) optimal arm.

*Basis:* for all the ends of time steps $t \in \{1, .., KM\}$ this holds trivially since, by the definition of SWA we pull each arm exactly $M$ times.

*Inductive hypothesis:* Assume that the above statement holds for the end of time step $t'$ such that, $KM \le t' < T$.

*Inductive step:* We show that the above statement holds for the end of time step $t' + 1$. By the non-increasing Assumption 2.1 we note two things: (1) The RHS of the inner inequality in Eq. (17) is non-increasing in $t$, thus if the inequality did not hold for some arm $j$ at the end of time step $t'$ it can only hold for it at the end of $t' + 1$ if SWA pulls arm $j$ in that round. (2) The number of $n$s for which the inequality holds for some arm $j$ can increase only by one at each time step. Combining the two with our inductive hypothesis we simply need to show that if for some arm $j$, Eq. (17) holds with equality (i.e., the number of $n$s is $M$), that arm would not be pulled in $t' + 1$. By the non-increasing Assumption 2.1 we know that the last $M$ expected rewards of arm $j$ are those who are at least $2q$ lower. Let $i^* \in \operatorname{argmax}_{i \in [K]} \left[ \mu_i \left( N_i^{\pi^{\text{SWA}}}(t' + 1) \right) \right]$ (if this set contains more than one arm, choose arbitrarily). We have,

$$\frac{1}{M} \sum_{n=N_j^{\pi^{\text{SWA}}}(t'+1)-M+1}^{N_j^{\pi^{\text{SWA}}}(t'+1)} r_j^n \overset{(1)}{<} \mathbb{E}\left[ \frac{1}{M} \sum_{n=N_j^{\pi^{\text{SWA}}}(t'+1)-M+1}^{N_j^{\pi^{\text{SWA}}}(t'+1)} r_j^n \right] + q \overset{(2)}{\le}$$

$$\mu_j \left( N_j^{\pi^{\text{SWA}}}(t'+1) - M + 1 \right) + q \overset{(3)}{<} \mu_{i^*} \left( N_{i^*}^{\pi^{\text{SWA}}}(t'+1) \right) - q \overset{(4)}{\le}$$

$$\mathbb{E}\left[ \frac{1}{M} \sum_{n=N_{i^*}^{\pi^{\text{SWA}}}(t'+1)-M+1}^{N_{i^*}^{\pi^{\text{SWA}}}(t'+1)} r_{i^*}^n \right] - q \overset{(5)}{<} \frac{1}{M} \sum_{n=N_{i^*}^{\pi^{\text{SWA}}}(t'+1)-M+1}^{N_{i^*}^{\pi^{\text{SWA}}}(t'+1)} r_{i^*}^n \qquad (18)$$

where (1) and (5) hold by our assumption regarding $|\bar{X}_M - \mathbb{E}[\bar{X}_M]| < q$, (2) and (4) hold by the non-increasing Assumption 2.1, and (3) holds by the definition of the inequality in Eq. (17). Since the SWA algorithm chooses in the Balance step according to the empirical averages of the last $M$-pulls of each arm, we infer that arm $j$ would not be pulled ($i^*$ has higher empirical average). This concludes the inductive step proof, and hence our statement holds.

**Step 2:** bound $J\left(T; \pi^{\widehat{\max}}\right) - J\left(T; \pi^{\text{SWA}}\right)$.

Let $\pi^{\widehat{\max}}$ be a policy defined by,

$$\pi^{\widehat{\max}}(t) \in \operatorname*{argmax}_{i \in [K]} \{\mu_i\left(N_i(t)\right)\} \qquad (19)$$

where we first pull each arm once using Round-Robin (before following the above rule), and in a case of tie, break it using the smallest index.

Define

$$I^{\widehat{\max}}(T) = \left\{ \left( i_t^{\widehat{\max}}, n_t^{\widehat{\max}} \right) \right\}_{t=1}^T \qquad (20)$$

to be the (deterministic) set of tuples induced by applying $\pi^{\widehat{\max}}$, where $i_t^{\widehat{\max}}$ is the arm chosen at time step $t$, and $n_t^{\widehat{\max}}$ is the time it is being pulled. In the same manner, we define the (stochastic) set $I^{\text{SWA}}(T)$, composed of $\left( i_t^{\text{SWA}}, n_t^{\text{SWA}} \right)$ tuples, which induced by applying $\pi^{\text{SWA}}$. We further define $I_{\setminus \text{SWA}}^{\widehat{\max}}(T) = I^{\widehat{\max}}(T) \setminus \{I^{\widehat{\max}}(T) \cap I^{\text{SWA}}(T)\}$, and $I_{\setminus \widehat{\max}}^{\text{SWA}}(T) = I^{\text{SWA}}(T) \setminus \{I^{\widehat{\max}}(T) \cap I^{\text{SWA}}(T)\}$,

and also $\mu_{\max}^{\text{SWA}}(T+1) = \max_{i \in [K]}\left[\mu_i\left(N_i^{\pi^{\text{SWA}}}(T+1)\right)\right]$. By Observation 2, the difference in the policies expected total rewards only depends on these number of pull sets. Since both policies start with one Round-Robin pulls of the arms we have,

$$
\begin{aligned}
J\left(T; \pi^{\widehat{\max}}\right) - J\left(T; \pi^{\text{SWA}}\right) &= \sum_{\left(i_t^{\widehat{\max}}, n_t^{\widehat{\max}}\right) \in I^{\widehat{\max}}} \mu_{i_t^{\widehat{\max}}}\left(n_t^{\widehat{\max}}\right) - \sum_{\left(i_t^{\text{SWA}}, n_t^{\text{SWA}}\right) \in I^{\text{SWA}}} \mu_{i_t^{\text{SWA}}}\left(n_t^{\text{SWA}}\right) \\
&= \sum_{\left(i_t^{\widehat{\max}}, n_t^{\widehat{\max}}\right) \in I_{\backslash \text{SWA}}^{\widehat{\max}}} \mu_{i_t^{\widehat{\max}}}\left(n_t^{\widehat{\max}}\right) - \sum_{\left(i_t^{\text{SWA}}, n_t^{\text{SWA}}\right) \in I_{\backslash \widehat{\max}}^{\text{SWA}}} \mu_{i_t^{\text{SWA}}}\left(n_t^{\text{SWA}}\right) \\
&\leq \mu_{\max}^{\text{SWA}}(T+1) \times |I_{\backslash \text{SWA}}^{\widehat{\max}}| - 0 \times KM \\
&\quad - \left(\mu_{\max}^{\text{SWA}}(T+1) - 2q\right) \times \left(|I_{\backslash \text{SWA}}^{\widehat{\max}}| - KM\right) \\
&\leq KM \max_{i \in [K]} \mu_i(1) + 2qT
\end{aligned}
\tag{21}
$$

The first inequality holds by: (1) the non-increasing Assumption 2.1 implies that all the tuples in $I_{\backslash \text{SWA}}^{\widehat{\max}}$ correspond to expected reward upper bounded by $\mu_{\max}^{\text{SWA}}(T+1)$, and (2) by what we showed in *Step 1*, there are at most $KM$ members in $I_{\backslash \widehat{\max}}^{\text{SWA}}$ that are more than $2q$ below $\mu_{\max}^{\text{SWA}}(T+1)$, and the positiveness of the expected rewards by Assumption 2.1. The second inequality holds by trivially bounding $\mu_{\max}^{\text{SWA}}(T+1) \leq \max_{i \in [K]} \mu_i(1)$, and $|I_{\backslash \text{SWA}}^{\widehat{\max}}| = |I_{\backslash \widehat{\max}}^{\text{SWA}}| \leq T$.

Finally, we note that all the above analysis was done assuming that $|\bar{X}_M - \mathbb{E}\left[\bar{X}_M\right]| < q$ for all arms throughout the trajectory, and we now address the case where it is violated. By Observation 1, the probability of the inequality to be violated $\leq 1/T^2$. The number of times this inequality is tested throughout the trajectory is bounded by $KT$ (for each of the arms, in every time step, during the Balance step), and if the inequality is violated (even once) then $J\left(T; \pi^{\widehat{\max}}\right) - J\left(T; \pi^{\text{SWA}}\right)$ is trivially bounded by $T \max_{i \in [K]} \mu_i(1)$ according to the non-increasing Assumption 2.1. Thus, we infer that in expectation we have,

$$
J\left(T; \pi^{\widehat{\max}}\right) - J\left(T; \pi^{\text{SWA}}\right) \leq KM \max_{i \in [K]} \mu_i(1) + 2qT + K \max_{i \in [K]} \mu_i(1)
\tag{22}
$$

***Step 3:*** bound the regret.

We bound the regret using our previous obtained result for $\pi^{\widehat{\max}}$ by,

$$
\begin{aligned}
\mathcal{R}\left(T; \pi^{\text{SWA}}\right) &= \max_{\pi \in \Pi}\{J(T; \pi)\} - J\left(T; \pi^{\text{SWA}}\right) \\
&= J(T; \pi^{\max}) - J\left(T; \pi^{\text{SWA}}\right) \\
&= \left(J(T; \pi^{\max}) - J\left(T; \pi^{\widehat{\max}}\right)\right) + \left(J\left(T; \pi^{\widehat{\max}}\right) - J\left(T; \pi^{\text{SWA}}\right)\right) \\
&\leq K \max_{i \in [K]} \mu_i(1) + \left(J\left(T; \pi^{\widehat{\max}}\right) - J\left(T; \pi^{\text{SWA}}\right)\right) \\
&\leq 2K \max_{i \in [K]} \mu_i(1) + KM \max_{i \in [K]} \mu_i(1) + 2qT \\
&= \left(\alpha \max_{i \in [K]} \mu_i(1) + \alpha^{-1/2}\right) 4^{2/3} \sigma^{2/3} K^{1/3} T^{2/3} \ln^{1/3}\left(\sqrt{2}T\right) + 3K \max_{i \in [K]} \mu_i(1)
\end{aligned}
\tag{23}
$$

where the first equality holds by Lemma 2.1, the first inequality holds by Theorem 3 in Heidari et al. [2016], the second inequality holds by the bound we found in *Step 2*, and the last equality holds by plugging in the definition for $M$ and $q$. This establishes Theorem 3.1.

### C.2 Proof of Corollary 3.1.1

For convenience, we define the following objects: $\mathcal{R}(t_1 \to t_2; \pi)$ is the regret accumulated between time steps $t_1$ and $t_2$ (included), by applying policy $\pi$ consistently. $\mathcal{R}(t_1 \to t_2; \pi_2|\pi_1(t_1))$ is the regret accumulated between time steps $t_1$ and $t_2$, by applying $\pi_1$ until time step $t_1$, and then $\pi_2$ for the measured time steps. We define similar objects for the expected total reward, $J$.

We note that,

$$J\left(t_1 \to t_2; \pi^{\max}\right) \leq J\left(t_1 \to t_2; \pi^{\max} \middle| \pi\left(t_1\right)\right), \quad \forall \pi \in \Pi \tag{24}$$

The above inequality can be understood by the following argument: consider a decreasing sorted list of all the expected rewards across all arms. By Assumption 2.1, at each time step, $\pi^{\max}$ simply pulls an arm corresponding to the highest element in that list, that was not previously pulled (independently of previous pulls).

Thus, $J\left(t_1 \to t_2; \pi^{\max}\right)$ is the sum of the $t_1^{\text{th}}$ to $t_2^{\text{th}}$ elements in this list, which is the lowest possible sum of the $|t_2 - t_1 + 1|$ highest elements in the list, following any $|t_1 - 1|$ pulls.

Consider the $n^{\text{th}}$ iteration of wSWA. i.e., between time steps $t_1 = 2^{n-1}$ and $t_2 = \min\left[2^n - 1, T\right]$. We have,

$$
\begin{aligned}
\mathcal{R}\left(t_1 \to t_2; \pi^{\text{wSWA}}\right) &\overset{(1)}{=} J\left(t_1 \to t_2; \pi^{\max}\right) - J\left(t_1 \to t_2; \pi^{\text{wSWA}}\right) \\
&\overset{(2)}{=} J\left(t_1 \to t_2; \pi^{\max} \middle| \pi^{\max}\left(t_1\right)\right) - J\left(t_1 \to t_2; \pi^{\text{wSWA}} \middle| \pi^{\text{wSWA}}\left(t_1\right)\right) \\
&\overset{(3)}{\leq} J\left(t_1 \to t_2; \pi^{\max} \middle| \pi^{\text{wSWA}}\left(t_1\right)\right) - J\left(t_1 \to t_2; \pi^{\text{wSWA}} \middle| \pi^{\text{wSWA}}\left(t_1\right)\right) \\
&\overset{(4)}{=} J\left(t_1 \to t_2; \pi^{\max} \middle| \pi^{\text{wSWA}}\left(t_1\right)\right) - J\left(t_1 \to t_2; \pi^{\text{SWA}} \middle| \pi^{\text{wSWA}}\left(t_1\right)\right) \\
&\overset{(5)}{=} \mathcal{R}\left(t_1 \to t_2; \pi^{\text{SWA}} \middle| \pi^{\text{wSWA}}\left(t_1\right)\right) \\
&\overset{(6)}{\leq} \mathcal{R}_{\text{bound}}\left(t_2 - t_1 + 1\right)
\end{aligned} \tag{25}
$$

where (1) and (2) hold by definition. (3) holds by Eq. (24). (4) by noting the wSWA applies SWA between $t_1$ and $t_2$. (5) by definition. (6) by observing that it is the regret of a known horizon problem that holds Assumption 2.1, thus we can use the upper bound from Theorem 3.1, denoted by $\mathcal{R}_{\text{bound}}$.

Let $\tilde{n} = \lfloor \log_2 T \rfloor + 1$, thus $2^{\tilde{n}-1} \leq T \leq 2^{\tilde{n}} - 1$, and we have,

$$
\begin{aligned}
\mathcal{R}\left(T; \pi^{\text{wSWA}}\right) &\overset{(1)}{=} \sum_{y=1}^{\tilde{n}-1} \mathcal{R}\left(2^{y-1} \to 2^y - 1; \pi^{\text{wSWA}}\right) + \mathcal{R}\left(2^{\tilde{n}-1} \to T; \pi^{\text{wSWA}}\right) \\
&\overset{(2)}{\leq} \sum_{y=1}^{\tilde{n}-1} \mathcal{R}_{\text{bound}}\left(2^{y-1}\right) + \mathcal{R}_{bound}\left(T - 2^{\tilde{n}-1} + 1\right) \\
&\overset{(3)}{\leq} \sum_{y=0}^{\tilde{n}-1} \mathcal{R}_{\text{bound}}\left(2^y\right) \\
&\overset{(4)}{=} \sum_{y=0}^{\tilde{n}-1} \left[A 2^{2y/3} \ln^{1/3}\left(2^{y+1/2}\right) + B\right] \\
&\overset{(5)}{\leq} A \ln^{1/3}\left(\sqrt{2}T\right) \sum_{y=0}^{\tilde{n}-1} 2^{2y/3} + B\left(\log_2 T + 1\right) \\
&\overset{(6)}{\leq} A 2^{5/3} T^{2/3} \ln^{1/3}\left(\sqrt{2}T\right) + B\left(\log_2 T + 1\right)
\end{aligned} \tag{26}
$$

where (1) holds by dividing the horizon and noting that the regret is additive. (2) holds by Eq (25). (3) holds by noting that both Theorem 3 from Heidari et al. and *Step 1* from the proof of Theorem 3.1 hold for any $t \in [T]$, thus the upper bound $\mathcal{R}_{\text{bound}}$ holds for any $t \in [T]$ (clearly, by plugging $T$ in the bound). (4) holds by plugging $\mathcal{R}_{\text{bound}}$ and defining $A = \left(\alpha \max_{i \in [K]} \mu_i\left(1\right) + \alpha^{-1/2}\right) 4^{2/3} \sigma^{2/3} K^{1/3}$, and $B = 3K \max_{i \in [K]} \mu_i\left(1\right)$. (5) holds by monotonicity of the logarithm, and noting that $A$ and $B$ are independent of $y$. Finally, (6) holds as a sum of a geometric series, and simple algebra.

Plugging back $A$ and $B$, we establish Corollary 3.1.1.

# D Parametric Case

## D.1 Proof of Thm. 4.1

**Bounding number of steps to optimality**
We first characterize the bound, and later show feasibility (i.e., that the analysis we show here indeed holds within the horizon).

Similar to the definition of $m^*_{\text{diff}}(p; \theta^*_i)$ and $m^*_{\text{diff}}(p)$, we define $m^*(p; \theta^*_i)$ as the solution to optimization problem (11) using Eq. (7) as the proximity rule to hypothesize $\hat{\theta}$, and $m^*(p) = \max_{\theta \in \Theta} m^*(p; \theta)$.

Let $T$ be some *unknown* horizon. We first show that $m^*\left(\frac{1}{KT^2}\right)$ is finite. Define,

$$\theta'_i(\tilde{m}) = \underset{\theta \neq \theta^*_i}{\operatorname{argmin}} \left\{ \left| \sum_{j=1}^{\tilde{m}} \mu(j; \theta^*_i) - \sum_{j=1}^{\tilde{m}} \mu(j; \theta) \right| \right\} \tag{27}$$

Thus we have, when we sample only from arm $i$,

$$P\left(\hat{\theta}_i(\tilde{m}) \neq \theta^*_i\right) = P\left(\exists \theta \neq \theta^*_i : |Y(i, \tilde{m}; \theta)| \leq |Y(i, \tilde{m}; \theta^*_i)|\right)$$

$$\leq P\left( \left| \sum_{j=1}^{\tilde{m}} r^i_j - \sum_{j=1}^{\tilde{m}} \mu(j; \theta^*_i) \right| > \frac{1}{2} \left| \sum_{j=1}^{\tilde{m}} \mu(j; \theta^*_i) - \sum_{j=1}^{\tilde{m}} \mu\left(j; \theta'_i(\tilde{m})\right) \right| \right)$$

$$\leq 2 \exp\left\{ -\frac{1}{8 \times det_{\theta^*_i, \theta'_i(\tilde{m})}(\tilde{m})} \right\}$$

(28)

where the first inequality holds by inclusion of events, and the second inequality holds by Eq. (14) and the definition of $det_{\theta^*_i, \theta'_i}$.

Since trivially $bal(n) \geq n$, by assumption 4.2, there exists a finite $\tilde{m}$, for which,

$$\max_{\theta_1 \neq \theta_2 \in \Theta^2} \left\{ det_{\theta_1, \theta_2}(\tilde{m}) \right\} \leq \frac{1}{8} \ln^{-1}\left(2KT^2\right) \tag{29}$$

Therefore, if we plug $\tilde{m}$ back in to the above equation we get,

$$2 \exp\left\{ -\frac{1}{8 \times det_{\theta^*_i, \theta'_i}(\bar{m})} \right\} \leq \frac{1}{KT^2} \tag{30}$$

Thus, we have a finite $\tilde{m}$ that satisfies the constraints of optimization problem (11) for $p = 1/KT^2$, and by definition $m^*\left(\frac{1}{KT^2}\right) \leq \tilde{m}$. i.e., $m^*\left(\frac{1}{KT^2}\right)$ is finite.

Given a rotting model, $\theta^*_i$ of arm $i$, we term that arm 'saturated' if it has been pulled at least $m^*\left(\frac{1}{KT^2}; \theta^*_i\right)$ times, which is finite since, by definition, $m^*\left(\frac{1}{KT^2}; \theta^*_i\right) \leq m^*\left(\frac{1}{KT^2}\right)$. We assume that once an arm is 'saturated', it is truely detected every time step, and omit this assertion from now on (we deal with the misdetection case later). i.e., we assume that once arm $i$ hypothesize its rotting model to be $\hat{\theta}_i$ and also has been pulled at least $m^*\left(\frac{1}{KT^2}; \theta^*_i\right)$ times, then $\hat{\theta}_i = \theta^*_i$.

We next bound the number of pulls of different arms, given the number of pulls of some other arm. Let $s$ be the first time step for which $\min_{i \in [K]}\{N_i(s)\} = \max_{\theta \in \Theta^*}\{m^*\left(\frac{1}{KT^2}; \theta\right)\}$. We first note that $s$ is finite since by Assumption 4.1 we have $\mu(n; \theta) \in o(1)$, combined with the $\operatorname{argmax}$ rule $\text{CTO}_{\text{SIM}}$ follows and its tie breaking rule, at some finite time step all arms would be pulled the specified amount of times. By our above assumption, from this point on, all the arms' rotting models are correctly detected. Thus, for any arm $j$, $N_j(s)$ can be upper bounded by the solution for,

$$\min t_j$$
$$\text{s.t} \begin{cases} t_j \in \mathbb{N} \\ t_j \geq \max_{\theta \in \Theta^*}\{m^*\left(\frac{1}{KT^2}; \theta\right)\} \\ \mu(t_j + 1; \theta^*_j) \leq \min_{\tilde{\theta} \in \Theta}\left[\mu\left(\max_{\theta \in \Theta^*}\left\{m^*\left(\frac{1}{KT^2}; \theta\right)\right\}; \tilde{\theta}\right)\right] \end{cases} \tag{31}$$

where the above optimization bound characterization holds since:

(1) For any arm $j \in \mathrm{argmin}_{i \in [K]}\{N_i(s)\}$, this holds trivially by the explicit constraint $t_j \geq \max_{\theta \in \Theta^*}\{m^*\left(\frac{1}{KT^2};\theta\right)\}$.

(2) For any arm $j \notin \mathrm{argmin}_{i \in [K]}\{N_i(s)\}$, clearly the constraint on the lower bound holds. As for the constraint on the upper bound, it holds by noting that all the arms' hypothesized models are correct and $\mathrm{CTO_{SIM}}$ follows an $\mathrm{argmax}$ policy, thus $j$ would not be pulled such that $\mu\left(N_j(s);\theta_j^*\right) < \min_{\theta \in \Theta}\left[\mu\left(\max_{\theta \in \Theta^*}\{m^*\left(\frac{1}{KT^2};\theta\right)\}\right)\right]$, as the RHS is the lowest obtainable expected reward until time step $s$. In addition, since the tie breaking rule is least # of pulls, its expected reward would not be equal to $\min_{\theta \in \Theta}\left[\mu\left(\max_{\theta \in \Theta^*}\{m^*\left(\frac{1}{KT^2};\theta\right)\}\right)\right]$.

Let $\mu_{min}(s;\Theta^*) = \min_{j \in [K]}\{\mu\left(N_j(s);\theta_j^*\right)\}$. Following $\mathrm{CTO_{SIM}}$ policy we infer that there exists $\tilde{s} \geq s$ for which:

(1) $\mu\left(N_i(\tilde{s})+1;\theta_i^*\right) \leq \mu_{min}(s;\Theta^*)$, for all $i \in [K]$.

(2) $\mu\left(N_i(\tilde{s});\theta_i^*\right) > \mu_{min}(s;\Theta^*)$, for all $i \notin \mathrm{argmin}_{j \in [K]}\{\mu\left(N_j(s);\theta_j^*\right)\}$.

The above observation holds by noting that $\mathrm{CTO_{SIM}}$ follows an $\mathrm{argmax}$ rule, thus it would choose arms $\notin \mathrm{argmin}_{j[K]}\{\mu\left(N_j(s);\theta_j^*\right)\}$ to be pulled as long as their expected reward is strictly greater than already pulled minimal expected reward $\mu_{min}(s;\Theta^*)$, before the possibility of choosing arms with expected reward $\leq \mu_{min}(s;\Theta^*)$. Since by Eq. (31) we have that $\min_{j \in [K]}\{\mu\left(N_j(s);\theta_j^*\right)\} \geq \min_{\tilde{\theta} \in \Theta}\left[\mu\left(\max_{\theta \in \Theta^*}\{m^*\left(\frac{1}{KT^2};\theta\right)\};\tilde{\theta}\right)\right]$, we can upper bound $\tilde{s}$ by the following,

$$
\begin{aligned}
& \min \|t\|_1 \\
& \text{s.t} \begin{cases} t \in \mathbb{N}^K \\ t_i \geq \max_{\theta \in \Theta^*}\{m^*\left(\frac{1}{KT^2};\theta\right)\}, \quad \forall i \in [K] \\ \mu\left(t_i+1;\theta_i^*\right) \leq \min_{\tilde{\theta} \in \Theta}\left[\mu\left(\max_{\theta \in \Theta^*}\left\{m^*\left(\frac{1}{KT^2};\theta\right)\right\};\tilde{\theta}\right)\right], \quad \forall i \in [K] \end{cases}
\end{aligned}
\tag{32}
$$

We turn to show optimality starting from time step $\tilde{s}$. We start by showing for $\tilde{s}$.

Assume on the contrary that, $J\left(\tilde{s};\pi^{\max}\right) \neq J\left(\tilde{s};\pi^{\mathrm{CTO_{SIM}}}\right)$. On the one hand, by Lemma 2.1, we have, $J\left(\tilde{s};\pi^{\max}\right) \geq J\left(\tilde{s};\pi^{\mathrm{CTO_{SIM}}}\right)$. On the other hand, Let $\{q_i\}_{i \in [K]}$ be the set of the arms' number of pulls at time $\tilde{s}$ following $\pi^{\max}$ (respectively, $\{\tilde{s}_i\}_{i \in [K]}$ for $\mathrm{CTO_{SIM}}$), i.e.,

$$
J\left(\tilde{s};\pi^{\max}\right) = \sum_{i \in [K]}\sum_{j=1}^{q_i}\mu(j;\theta_i^*)
\tag{33}
$$

We have that $J\left(\tilde{s};\pi^{\mathrm{CTO_{SIM}}}\right) - J\left(\tilde{s};\pi^{\max}\right)$ is a sum of pairs in the form of, $\mu\left(l;\theta_i^*\right) - \mu\left(h;\theta_j^*\right)$ where $l \leq \tilde{s}_i$, and $h > \tilde{s}_j$, for $i \neq j \in [K]$. By definition of $\{\tilde{s}_i\}$ and the non-increasing assumption 2.1, we have that $\mu\left(l;\theta_i^*\right) \geq \mu_{min}(s;\Theta^*)$, and $\mu_{min}(s;\Theta^*) \geq \mu\left(h;\theta_j^*\right)$, resulting in $J\left(\tilde{s};\pi^{\mathrm{CTO_{SIM}}}\right) \geq J\left(\tilde{s};\pi^{\max}\right)$. Hence, the regret vanishes in time step $\tilde{s}$, achieving optimality.

We next show that the regret remains zero for $\hat{s} \geq \tilde{s}$.

We showed optimality for time step $\tilde{s}$ defined above. We next show optimality for $\tilde{s}+1$. We examine the two possible cases.

_Case 1_: $\forall i \in [K] : q_i = \tilde{s}_i$. Since $\mathrm{CTO_{SIM}}$ follows the $\mathrm{argmax}$ rule as $\pi^{\max}$ does, we infer that arms with equal expected reward would be chosen by both $\mathrm{CTO_{SIM}}$ and $\pi^{\max}$. Thereby, holding $J\left(\tilde{s}+1;\pi^{\max}\right) = J\left(\tilde{s}+1;\pi^{\mathrm{CTO_{SIM}}}\right)$. i.e., zero regret as stated.

_Case 2_: $\exists i : \tilde{s}_i \neq q_i$. Therefore, there is an arm, denoted as $i_{gap}$, for which $\tilde{s}_{i_{gap}} < q_{i_{gap}}$. By the $\mathrm{argmax}$ rule, $\mathrm{CTO_{SIM}}$ chooses an arm $i_{\tilde{s}+1}$ such that, $\mu\left(\tilde{s}_{i_{\tilde{s}+1}}+1;\theta_{i_{\tilde{s}+1}}^*\right) \geq \mu\left(\tilde{s}_{i_{gap}}+1;\theta_{i_{gap}}^*\right)$. By the non-increasing assumption 2.1, and the definition of $\pi^{\max}$, since $q_{i_{gap}} \geq \tilde{s}_{i_{gap}}+1$, we have $\mu\left(q_{j_{\tilde{s}+1}};\theta_{j_{\tilde{s}+1}}^*\right) \leq \mu\left(q_{i_{gap}};\theta_{i_{gap}}^*\right) \leq \mu\left(\tilde{s}_{i_{gap}}+1;\theta_{i_{gap}}^*\right)$, where $j_{\tilde{s}+1}$ is the arm chosen by $\pi^{\max}$. Thus, on the one hand we have $J\left(\tilde{s}+1;\pi^{\max}\right) \leq J\left(\tilde{s}+1;\pi^{\mathrm{CTO_{SIM}}}\right)$. On the other hand, by Lemma 2.1, we have $J\left(\tilde{s}+1;\pi^{\max}\right) \geq J\left(\tilde{s}+1;\pi^{\mathrm{CTO_{SIM}}}\right)$. Combining the two, we have $J\left(\tilde{s}+1;\pi^{\max}\right) = J\left(\tilde{s}+1;\pi^{\mathrm{CTO_{SIM}}}\right)$. i.e., zero regret as stated.

The above argument can be applied recursively for any $\hat{s} > \tilde{s}$, thus establishing optimality of $\mathrm{CTO_{SIM}}$ for all $\hat{s} \geq s$, under true detection.

If it happens to be that $\|t\|_1 \leq T$, then for that $T$, $\text{CTO}_{\text{SIM}}$ will achieve zero regret (starting from $\tilde{s}$). Since we require that the result will hold from some $T^*_{\text{SIM}}$ onward, we need the above characterization to also hold for any $\tilde{T} \geq T$. We thereby infer that the smallest $T$ such that for any $\tilde{T} \geq T$, there exists $t$ for which the above stated result holds (i.e., the solution to the optimization problem is indeed holds $\|t\|_1 \leq \tilde{T}$), can serve as an upper bound for $T^*_{\text{SIM}}$, resulting in $T^*_{\text{SIM}}$ being upper bounded by the solution for,

$$\min T$$

$$\text{s.t} \begin{cases} T, b \in \mathbb{N} \cup \{0\}, t \in \mathbb{N}^K \\ \forall b, \exists t : \begin{cases} \|t\|_1 \leq T + b \\ t_i \geq \max_{\theta \in \Theta^*} \left\{ m^* \left( \frac{1}{K(T+b)^2}; \theta \right) \right\} \\ \mu\left(t_i + 1; \theta_i^*\right) \leq \min_{\tilde{\theta} \in \Theta} \left[ \mu\left( \max_{\theta \in \Theta^*} \left\{ m^* \left( \frac{1}{K(T+b)^2}; \theta \right) \right\}; \tilde{\theta} \right) \right] \end{cases} \end{cases} \tag{34}$$

**Feasibility**

In order to show feasibility, we wish to obtain,

$$\{\text{\# of steps for Detection}\} + \{\text{\# of steps for Balance}\} \leq T$$

where Detection is a phase of pulling arms until the rotting models are detected with high enough probability (defined below), and Balance is a phase which at the end of it there is no arm which yields strictly higher expected reward than the minimal observed expected reward so far, as explained in the former step, resulting in vanishing regret (similar to $s$ and $\tilde{s}$ discussed above). We require that the detection of each arm is w.p of at least $1 - \frac{1}{KT^2}$. Define $W(T) = \max_{\theta_1, \theta_2} \left\{ det^{\star \downarrow}_{\theta_1, \theta_2} \left( \frac{1}{16} \ln^{-1} \left( \sqrt{2K}T \right) \right) \right\}$. As shown in the beginning of this proof, after pulling an arm for $W(T)$ times, the probability of misdetection its rotting model $\leq \frac{1}{KT^2}$. We refer to an arm that has been pulled at least $W(T)$ times as 'strongly saturated'. From now on we will assume that any 'strongly saturated' arm is truely detected at each decision point, and will discuss the other case later on.

On the one hand, by the definition of $bal()$, the non-increasing assumption 2.1, and the rule of tie breaking applied by $\text{CTO}_{\text{SIM}}$, we have that all arms become 'strongly saturated' after, at most, $W(T) + (K - 1) \times bal(W(T))$ time steps.

On the other hand, from the definition of $bal()$, and $\text{CTO}_{\text{SIM}}$, we infer that no arm would be pulled $bal(W(T)) + 1$ times before all other arms would become 'strongly saturated'.

Combining the two above observations we have that, after at most $W(T) + (K - 1) \times bal(W(T))$ time steps, there exists a time step in which all arms have became 'strongly saturated', but were not pulled more than $bal(W(T))$ times. From that point, following the same flow at the former subsection, the total number of pulls required in order to "balance" the arms (i.e., there is no pull that would yield strictly higher reward than the minimal expected reward observed so far), is bounded by $K \times bal(W(T))$. That is under the worst case scenario, where every arm that becomes 'strongly saturated' is detected to be an arm that requires $bal(W(T))$ pulls to "balance" itself w.r.t to another 'strongly saturated' arm. Thus, we infer that,

$$\{\text{\# of steps for Detection}\} + \{\text{\# of steps for Balance}\} \leq K \times bal(W(T))$$

Let $\epsilon = \left( K\sqrt{2K} \right)^{-1}$. By assumption 4.2, we have that there exists a finite $\tilde{T}_{max}$ for which,

$$\forall \tilde{T} \geq \tilde{T}_{max} : bal\left( \max_{\theta_1 \neq \theta_2 \in \Theta^2} \left\{ det^{\star \downarrow}_{\theta_1, \theta_2} \left( \frac{1}{16} \ln^{-1} \left( \tilde{T} \right) \right) \right\} \right) \leq \epsilon \tilde{T} \tag{35}$$

We denote $T = \left( \sqrt{2K} \right)^{-1} \tilde{T}$, and get,

$$\forall T \geq \frac{\tilde{T}_{max}}{\sqrt{2K}} : K \times bal(W(T)) \leq T \tag{36}$$

which implies, under true detection, that $\forall T \geq \tilde{T}_{max}/\sqrt{2K}$, $\text{CTO}_{\text{SIM}}$ algorithm achieves zero regret. Since by definition we have $\forall \theta \in \Theta : m^* \left( \frac{1}{KT^2}; \theta \right) \leq m^* \left( \frac{1}{KT^2} \right)$, and by definition of $m^* \left( \frac{1}{KT^2} \right)$

we have $m^* \left( \frac{1}{KT^2} \right) \leq W(T)$, we infer that there exists (a finite) $T_{\text{SIM}}^*$ that holds the optimization problem characterization as stated above (i.e., $\forall \tilde{T} \geq T_{\text{SIM}}^*$ the optimization problem is feasible).

**Misdetection and Expectation**

So far, we assumed that each 'saturated' (or 'strongly saturated') arm is truely detected. By definition each 'saturated' (or 'strongly saturated') arm probability of misdetection in any time step is upper bounded by $1/KT^2$. Thereby, after all the arms are 'saturated', the probability of a misdetection in each time step is upper bounded by $1/T^2$. The number of time steps where all the arms are 'saturated' (referred to as the 'saturated step') is trivially bounded by $T$. Hence, the probability that a misdetection occurs after the 'saturated step' is bounded by $1/T$. Meaning that $\forall T \geq T_{\text{SIM}}^*$, $\text{CTO}_{\text{SIM}}$ achieves zero regret w.p of at least $1 - 1/T$.

Next, we note that, as for the case where we misdetect any arm,

$$
\begin{aligned}
J\left(T; \pi^{\max}\right) - J\left(T; \pi^{\text{CTO}_{\text{SIM}}}\right) &= \sum_{i=1}^{K} \sum_{j=1}^{N_i^{\max}(T)} \mu\left(j; \theta_i^*\right) - \sum_{i=1}^{K} \sum_{j=1}^{N_i^{\text{CTO}_{\text{SIM}}}(T)} \mu\left(j; \theta_i^*\right) \\
&\leq \sum_{i=1}^{K} I_{\{N_i^{\max}(T) > N_i^{\text{CTO}_{\text{SIM}}}(T)\}} \sum_{N_i^{\text{CTO}_{\text{SIM}}}(T)+1}^{N_i^{\max}(T)} \mu\left(j; \theta_i^*\right) \\
&\leq T \max_{\theta \in \Theta^*} \left\{ \mu\left( \min_{i \in [K]} \{N_i^{\text{CTO}_{\text{SIM}}}(T)\}; \theta \right) \right\}
\end{aligned}
\tag{37}
$$

where the first inequality holds by only considering cases where $N_i^{\max}(T) > N_i^{\text{CTO}_{\text{SIM}}}(T)$, and not the other way around (since the expected rewards are positive by Assumption 4.1).

By applying expectation over events (true detection or not), we get,

$$
\begin{aligned}
\mathcal{R}\left(T; \pi^{\text{CTO}_{\text{SIM}}}\right) &= \mathcal{R}\left(T; \pi^{\text{CTO}_{\text{SIM}}} | \text{true detection}\right) \times P\left(\text{true detection}\right) \\
&\quad + \mathcal{R}\left(T; \pi^{\text{CTO}_{\text{SIM}}} | \text{misdetection}\right) \times P\left(\text{misdetection}\right) \\
&\leq \max_{\theta \in \Theta^*} \left\{ \mu\left( \min_{i \in [K]} \{N_i^{\text{CTO}_{\text{SIM}}}(T)\}; \theta \right) \right\}
\end{aligned}
\tag{38}
$$

Finally,

$$
\begin{aligned}
T &= \sum_{i=1}^{K} N_i^{\text{CTO}_{\text{SIM}}}(T) \\
&\leq \min_{i \in [K]} N_i^{\text{CTO}_{\text{SIM}}}(T) + (K-1) \max_{i \in [K]} N_i^{\text{CTO}_{\text{SIM}}}(T) \\
&\leq \min_{i \in [K]} N_i^{\text{CTO}_{\text{SIM}}}(T) + (K-1) \times bal\left( \min_{i \in [K]} N_i^{\text{CTO}_{\text{SIM}}}(T) \right) \\
&\leq K \times bal\left( \min_{i \in [K]} N_i^{\text{CTO}_{\text{SIM}}}(T) \right)
\end{aligned}
\tag{39}
$$

Hence, by assumption 2.1, $\min_{i \in [K]} N_i^{\text{CTO}_{\text{SIM}}}(T) \xrightarrow{T \to \infty} \infty$, resulting in $\mathcal{R}\left(T; \pi^{\text{CTO}_{\text{SIM}}}\right) \in o(1)$, and trivially $\leq \max_{\theta \in \Theta^*} \mu(1; \theta)$.

We **Note** that from the feasibility step, given a function $U(\epsilon)$ that satisfies $\forall n \geq U(\epsilon)$,

$$
bal\left( \max_{\theta_1 \neq \theta_2 \in \Theta^2} \left\{ det_{\theta_1, \theta_2}^{\star \downarrow}\left( \frac{1}{16} \ln^{-1}(n) \right) \right\} \right) \leq \epsilon n
\tag{40}
$$

we have,

$$
T_{\text{SIM}}^* \leq \frac{U\left( \left(K\sqrt{2K}\right)^{-1} \right)}{\sqrt{2K}}
\tag{41}
$$

## D.2 Proof of Thm. 4.2

**Decomposing the regret**
First, we upper bound the regret by,

$$
\mathcal{R}\left(T; \pi^{\text{D-CTO}_{\text{UCB}}}\right) = \sum_{i=1}^{K} \sum_{j=1}^{\mathbb{E}\left[N_i^{\pi^{\max}}(T)\right]} \mu_i\left(j\right) - \sum_{i=1}^{K} \sum_{j=1}^{\mathbb{E}\left[N_i^{\pi^{\text{D-CTO}_{\text{UCB}}}}(T)\right]} \mu_i\left(j\right)
$$

$$
\leq \underbrace{\sum_{i \neq a^*} \sum_{j=1}^{\mu^{\star\downarrow}(\Delta_i; \theta_i^*)} \mu_i\left(j\right)}_{=\tilde{C}\left(\Theta^*, \{\mu_i^c\}\right)} + \sum_{j=1}^{T} \mu_{a^*}\left(j\right) - \sum_{i=1}^{K} \sum_{j=1}^{\mathbb{E}\left[N_i^{\pi^{\text{D-CTO}_{\text{UCB}}}}(T)\right]} \mu_i\left(j\right)
$$

$$
= \tilde{C}\left(\Theta^*, \{\mu_i^c\}\right) + \sum_{\mathbb{E}\left[N_{a^*}^{\pi^{\max}}(T)\right]+1}^{T} \mu_{a^*}\left(j\right) - \sum_{i \neq a^*} \sum_{j=1}^{\mathbb{E}\left[N_i^{\pi^{\text{D-CTO}_{\text{UCB}}}}(T)\right]} \mu_i\left(j\right)
$$

$$
\leq \tilde{C}\left(\Theta^*, \{\mu_i^c\}\right) + \sum_{\mathbb{E}\left[N_{a^*}^{\pi^{\max}}(T)\right]+1}^{T} \left(\mu_{a^*}^c + \mu\left(1; \theta_{a^*}^*\right)\right) - \sum_{i \neq a^*} \sum_{j=1}^{\mathbb{E}\left[N_i^{\pi^{\text{D-CTO}_{\text{UCB}}}}(T)\right]} \mu_i^c
$$

$$
\leq \tilde{C}\left(\Theta^*, \{\mu_i^c\}\right) + \sum_{i \neq a^*} \mathbb{E}\left[N_i^{\pi^{\text{D-CTO}_{\text{UCB}}}}(T)\right] \times \left(\Delta_i + \mu\left(1; \theta_{a^*}^*\right)\right)
$$

$$(42)$$

where $\mathbb{E}\left[N_i^{\pi^{\max}}(T)\right]$ is the expected number of pulls of arm $i$ at time $T$ induced by the optimal policy, $\pi^{\max}$, and $\mathbb{E}N_i^{\pi^{\text{D-CTO}_{\text{UCB}}}}(T)$ is the expected number of pulls induced by policy $\pi^{\text{D-CTO}_{\text{UCB}}}$. The first inequality holds by noting that $\pi^{\max}$ pulls according to $\text{argmax}$ rule, thus any arm $i \neq a^*$ would not be pulled after yielding expected reward not greater than $\mu_{a^*}^c$, according to the behavior of $\mu\left(\cdot; \cdot\right)$ by assumption 2.1.

**Detecting the models**
Next, we show that $m_{\text{diff}}^*\left(\delta/K\right)$ is finite. Define,

$$
D\left(\mu\left(\cdot; \theta\right), 1, n\right) = \sum_{j=1}^{\lfloor \frac{n}{2} \rfloor} \mu\left(j; \theta\right) - \sum_{j=\lfloor \frac{n}{2} \rfloor + 1}^{n} \mu\left(j; \theta\right) \tag{43}
$$

and,

$$
\theta_i'\left(\tilde{m}\right) = \underset{\theta \neq \theta_i^*}{\text{argmin}} \left\{ \left| \mathcal{D}\left(\mu\left(\cdot; \theta_i^*\right), 1, \tilde{m}\right) - \mathcal{D}\left(\mu\left(\cdot; \theta\right), 1, \tilde{m}\right) \right| \right\} \tag{44}
$$

Thus, we have, when we sample only from arm $i$, and for an even $\tilde{m}$

$$
P\left(\hat{\theta}_i\left(\tilde{m}\right) \neq \theta_i^*\right) = P\left(\exists \theta \neq \theta_i^* : \left| Z\left(i, \tilde{m}; \theta\right) \right| \leq \left| Z\left(i, \tilde{m}; \theta_i^*\right) \right|\right)
$$

$$
\leq P\left(\left| \left(\sum_{j=1}^{\frac{\tilde{m}}{2}} r_j^i - \sum_{j=\frac{\tilde{m}}{2}+1}^{\tilde{m}} r_j^i\right) - \mathcal{D}\left(\mu\left(\cdot; \theta_i^*\right), 1, \tilde{m}\right) \right| >
$$

$$
\frac{1}{2} \left| \mathcal{D}\left(\mu\left(\cdot; \theta_i^*\right), 1, \tilde{m}\right) - \mathcal{D}\left(\mu\left(\cdot; \theta_i'\left(\tilde{m}\right)\right), 1, \tilde{m}\right) \right| \right)
$$

$$
\leq 2 \exp\left\{ - \frac{1}{8 \times Ddet_{\theta_i^*, \theta_i'(\tilde{m})}\left(\tilde{m}\right)} \right\}
$$

$$(45)$$

where the first inequality holds by inclusion of events, and the second inequality holds by Eq. (14), the definition of $Ddet_{\theta_i^*,\theta_i'}$, and noting that for an even $\tilde{m}$ we have,

$$\mathbb{E}\left[\sum_{j=1}^{\frac{\tilde{m}}{2}} r_j^i - \sum_{j=\frac{\tilde{m}}{2}+1}^{\tilde{m}} r_j^i\right] = \mathcal{D}\left(\mu\left(\cdot;\theta_i^*\right),1,\tilde{m}\right) \tag{46}$$

By assumption 4.3, there exists a finite, even, $\tilde{m}$ for which,

$$\max_{\theta_1\neq\theta_2\in\Theta^2}\left\{Ddet_{\theta_1,\theta_2}\left(\tilde{m}\right)\right\} \leq \frac{1}{8}\ln^{-1}\left(\frac{2K}{\delta}\right) \tag{47}$$

If we plug $\tilde{m}$ back to the above equation we get,

$$2\exp\left\{-\frac{1}{8\times Ddet_{\theta_i^*,\theta_i'(\tilde{m})}\left(\tilde{m}\right)}\right\} \leq \frac{\delta}{K} \tag{48}$$

Thus, we have a finite $\tilde{m}$ that satisfies the constraints of Prob. (11) for $p = \delta/K$, and by definition $m_{\text{diff}}^*\left(\delta/K\right) \leq \tilde{m}$. i.e., $m_{\text{diff}}^*\left(\delta/K\right)$ is finite.

**Bounding number of pulls**

We wish to bound $\mathbb{E}\left[N_i^{\pi^{\text{D-CTO}_{\text{UCB}}}}(T)\right]$ for all $i \neq a^*$. Remember that in the exploration part (leading to the Detect step), we pull each arm $m_{\text{diff}}^*\left(\delta/K\right)$ times, hence,

$$N_i^{\pi^{\text{D-CTO}_{\text{UCB}}}}(T) = m_{\text{diff}}^*\left(\delta/K\right) + \sum_{t=K\times m_{\text{diff}}^*(\delta/K)+1}^{T} 1_{\{i(t)=i\}} \tag{49}$$

where $1_{\{.\}}$ is the indicator function. Similarly to the proof of UCB1 (Auer et al. [2002a]) we have,

$$N_i^{\pi^{\text{D-CTO}_{\text{UCB}}}}(T) \leq l_i + \sum_{t=1}^{\infty}\sum_{s=m_{\text{diff}}^*(\delta/K)}^{t-1}\sum_{s_i=l_i}^{t-1} 1_{\{\hat{\mu}_{a^*}^c(s)+\mu(s;\theta_{a^*}^*)+c_{t,s}\leq\hat{\mu}_i^c(s_i)+\mu(s_i;\theta_i^*)+c_{t,s_i}\}} \tag{50}$$

where for some $\epsilon_i \in (0,\Delta_i)$, we denote $l_i = \max\left\{m_{\text{diff}}^*\left(\delta/K\right), \mu^{\star\downarrow}\left(\epsilon_i;\theta_i^*\right), \left\lceil\frac{32\sigma^2\ln T}{(\Delta_i-\epsilon_i)^2}\right\rceil\right\}$, and we note that we assume that we have detected the true underlying rotting models (holds w.p of at least $1-\delta$ as shown above).

The above indicator function holds when at least one of the following holds,

$$\begin{cases} \hat{\mu}_{a^*}^c(s) \leq \mu_{a^*}^c - c_{t,s} \\ \hat{\mu}_i^c(s_i) \geq \mu_i^c + c_{t,s_i} \\ \mu_{a^*}^c + \mu(s;\theta_{a^*}^*) < \mu_i^c + \mu(s_i;\theta_i^*) + 2c_{t,s_i} \end{cases} \tag{51}$$

Plugging $c_{t,s}$ and $c_{t,s_i}$, and using Eq. (14), we have,

$$\begin{cases} P\left(\hat{\mu}_{a^*}^c(s) \leq \mu_{a^*}^c - c_{t,s}\right) = t^{-4} \\ P\left(\hat{\mu}_i^c(s_i) \geq \mu_i^c + c_{t,s_i}\right) = t^{-4} \end{cases} \tag{52}$$

And for $s_i \geq l_i$ we have,

$$\begin{aligned} \mu_{a^*}^c + \mu(s;\theta_{a^*}^*) - \mu_i^c - \mu(s_i;\theta_i^*) - 2c_{t,s_i} &\geq \mu_{a^*}^c - \mu_i^c - \mu(s_i;\theta_i^*) - 2c_{t,s_i} \\ &\geq \mu_{a^*}^c - \mu_i^c - \epsilon_i - 2c_{t,s_i} \\ &= (\Delta_i - \epsilon_i) - 2c_{t,s_i} \\ &\geq 0 \end{aligned} \tag{53}$$

where the first inequality holds by assumption 4.1, the second inequality by $s_i \geq \mu^{\star\downarrow}\left(\epsilon_i;\theta_i^*\right)$, and the third inequality by $s_i \geq \left\lceil\frac{32\sigma^2\ln T}{(\Delta_i-\epsilon_i)^2}\right\rceil$.

Thus, combining the above observations, we get,

$$\begin{aligned} \mathbb{E}\left[N_i^\pi(T)\right] &\leq l_i + \sum_{t=1}^{\infty}\sum_{s=m_{\text{diff}}^*(\delta/K)}^{t-1}\sum_{s_i=l_i}^{t-1}\left(P\left(\hat{\mu}_{a^*}^c \leq \mu_{a^*}^c - c_{t,s}\right) + P\left(\hat{\mu}_i^c \geq \mu_i^c + c_{t,s_i}\right)\right) \\ &\leq l_i + \frac{\pi^2}{3} \end{aligned} \tag{54}$$

Denoting $C\left(\Theta^*,\{\mu_i^c\}\right) = \tilde{C}\left(\Theta^*,\{\mu_i^c\}\right) + \sum_{i\neq a^*}\frac{\pi^2+3}{3}\left(\Delta_i + \mu(1;\theta_{a^*}^*)\right)$, and plugging back into the upper bound on the regret, we achieve the stated result.

# E  Example 4.1

Next, we show an example for which the different assumptions hold; the case where the reward of arm $i$ for its $n^{\text{th}}$ pull is distributed as $\mathcal{N}\left(\mu_i^c + n^{-\theta_i^*}, \sigma^2\right)$. Where $\theta_i^* \in \Theta = \{\theta_1, \theta_2, ..., \theta_M\}$, and $\forall \theta \in \Theta : 0.01 \leq \theta \leq 0.49$.

## E.1  Assumption 4.1

The assumption given by $\mu(n; \theta)$ is positive, non-increasing in $n$, and $\mu(n; \theta) \in o(1), \forall \theta \in \Theta$, where $\Theta$ is a discrete known set. Indeed, for any $\theta \in \{\theta_1, \theta_2, ..., \theta_M\}$, which is a discrete known set where $0.01 \leq \theta \leq 0.49$, we have $n^{-\theta} \geq 0$ for all $n \geq 1$. Moreover, $\frac{\partial n^{-\theta}}{\partial \theta} = -\theta n^{-\theta-1} < 0$ for all $n \geq 1$, and $n^{-\theta} \xrightarrow{n\to\infty} 0$.

## E.2  Assumption 4.2

The assumption is given by,

$$bal\left(\max_{\theta_1 \neq \theta_2 \in \Theta^2}\left\{det_{\theta_1,\theta_2}^{\star\downarrow}\left(\frac{1}{16}\ln^{-1}(\zeta)\right)\right\}\right) \in o(\zeta) \tag{55}$$

Without a loss of generality, assume $\theta_2 > \theta_1$. We have for large enough $n$,

$$
\begin{aligned}
det_{\theta_1,\theta_2}(n) &= \frac{n\sigma^2}{\left(\sum_{j=1}^n j^{-\theta_1} - \sum_{j=1}^n j^{-\theta_2}\right)^2} \\
&\leq \frac{n\sigma^2}{\left(c_1 n^{1-\theta_1} - c_1 - c_2 n^{1-\theta_2}\right)^2} \\
&= \frac{n\sigma^2}{c_1^2 n^{2-2\theta_1} + c_2^2 n^{2-2\theta_2} - 2c_1 c_2 n^{2-\theta_1-\theta_2} - 2c_1^2 n^{1-\theta_1} + 2c_1 c_2 n^{1-\theta_2} + c_1^2} \\
&\leq \frac{n\sigma^2}{\tilde{c}n^{2-2\theta_1}} \\
&= \frac{\bar{c}}{n^{1-2\theta_1}}
\end{aligned}
\tag{56}
$$

where $\{c_1, c_2, \tilde{c}, \bar{c}\}$ are positive constants (independent of $n$). The first inequality holds by bounding the sums by integrals and keeping in mind that $\theta_2 > \theta_1$ combined with $0.01 \leq \theta \leq 0.49$. The second inequality holds from large enough $n$ (leading exponent, depends only on $\{\theta_1, \theta_2\}$, but finite). Next, we have,

$$\frac{\bar{c}}{n^{1-2\theta_1}} < \frac{1}{16}\ln^{-1}(\zeta) \implies n > (16\bar{c}\ln(\zeta))^{\frac{1}{1-2\theta_1}} > (16\bar{c}\ln(\zeta))^{50} \tag{57}$$

Meaning that $\zeta$ large enough,

$$\max_{\theta_1 \neq \theta_2 \in \Theta^2}\left\{det_{\theta_1,\theta_2}^{\star\downarrow}\left(\frac{1}{16}\ln^{-1}(\zeta)\right)\right\} < (16\bar{c}\ln(\zeta))^{50} \tag{58}$$

Next, we have,

$$\alpha^{-0.1} \leq x^{-0.49} \implies \alpha \geq x^{4.9} \tag{59}$$

Hence, $bal(x) = x^{4.9}$. Since $bal(\cdot)$ is monotonically increasing, we have that for $\zeta$ large enough,

$$bal\left(\max_{\theta_1 \neq \theta_2 \in \Theta^2}\left\{det_{\theta_1,\theta_2}^{\star\downarrow}\left(\frac{1}{16}\ln^{-1}(\zeta)\right)\right\}\right) < \hat{c}\ln^{245}(\zeta) \tag{60}$$

where $\hat{c}$ is a positive constant (independent of $\zeta$). Finally, we note that,

$$\lim_{\zeta \to \infty}\frac{\ln^{245}(\zeta)}{\zeta} = 0 \tag{61}$$

Thus we infer that the assumption holds.

## E.3 Assumption 4.3

The assumption is given by,

$$\max_{\theta_1 \neq \theta_2 \in \Theta^2} \left\{ Ddet_{\theta_1,\theta_2}^{\star\downarrow} (\epsilon) \right\} \leq B(\epsilon) < \infty, \quad \forall \epsilon > 0 \tag{62}$$

Without a loss of generality, assume $\theta_2 > \theta_1$. We have for large enough $n$,

$$
\begin{aligned}
Ddet_{\theta_1,\theta_2}(n) &= \frac{n\sigma^2}{\left( \left( \sum_{j=1}^{\lfloor \frac{n}{2} \rfloor} j^{-\theta_1} - \sum_{j=\lfloor \frac{n}{2} \rfloor +1}^{n} j^{-\theta_1} \right) - \left( \sum_{j=1}^{\lfloor \frac{n}{2} \rfloor} j^{-\theta_2} - \sum_{j=\lfloor \frac{n}{2} \rfloor +1}^{n} j^{-\theta_2} \right) \right)^2} \\
&\leq \frac{n\sigma^2}{\left( c_1 \left( -1 + 2\lfloor \frac{n}{2} \rfloor^{1-\theta_1} - n^{1-\theta_1} \right) - c_2 \left( 2 \left( \lfloor \frac{n}{2} \rfloor + 1 \right)^{1-\theta_2} - n^{1-\theta_2} \right) \right)^2} \\
&\leq \frac{n\sigma^2}{\tilde{c} n^{2-2\theta_1}} \\
&= \frac{\tilde{c}}{n^{1-2\theta_1}}
\end{aligned}
\tag{63}
$$

where $\{c_1, c_2, \tilde{c}\}$ are positive constants (independent of $n$). The inequalities hold by the same arguments as in E.2. Again, following the same logic as the end of E.2, we have that the assumption holds.