[Reviews · NeurIPS 2017]

Reviewer 1



This paper studies a kind of non-stationary stochastic bandits in which the expected reward of each arm decays as a function of the number of choosing it. They consider three cases: the reward function is (1) non-parametric, (2) parametric and decaying to 0, and (3) parametric and decaying to a positive constant that depends on each arm. For each case, they propose an algorithm and analyze its regret upper bound. They also conducted simulations which show superiority of their proposed algorithms compared with existing algorithms. Nicely written paper. The assumption of the rested rotting bandits seems natural in the applications of advertising and recommendation. The value of the proposed methods would increase by demonstrating effectiveness for real datasets using rotting models that are suited to the applications.

Reviewer 2



The paper introduces a novel variant of the multi-armed problem, motivated by practice. The approach is technically sound and well grounded. There are a few typos and minor concerns (listed below), despite which I believe this paper can be accepted. Decaying bandits - e.g., Komiyama and Qin 2014, Heidari, Kearns and Roth 2016 - are mentioned, but Deteriorating bandits - e.g., Gittins 1979, Mandelbaum 1987, Kaspi and Mandelbaum 1998, etc. - are not Can't you use algorithms from non-stationary MABs, which you say are "most related to our problem", for benchmarking? It's quite unfair to use only UCB variants. Line 15: "expanse" -> "expense" Line 19: you have skipped a number of key formulations and break-throughs between 1933 and 1985 Line 31-32: probably a word is missing in the sentence Line 63: "not including" what? Line 68: "equal objective" -> "equivalent objective" Line 68: this is an unusual definition of regret, in fact this expression is usually called "suboptimality", as you only compare to the optimal policy which is based on past allocations and observations. Regret is usually defined as distance from the objective achievable if one knew the unknown parameters. Line 105: in the non-vanishing case, do you allow \mu_i^c = 0 or negative? Line 106-118: some accompanying text describing the intuition behind these expressions would be helpful Line 119,144: headings AV and ANV would be more reasonable than names of the heuristics Line 160: "w.h.p" -> "w.h.p." Table 1: "CTO" is ambiguous, please use the full algorithm acronyms Figure 1: it would be desirable to explain the apparent non-monotonicity Line 256: It would be helpful for the reader if this sentence appeared in the Introduction. Btw, the term "rested bandits" is not very common; instead "classic bandits" is a much more popular term

Reviewer 3



This paper studies a variation of the classical multi-armed bandits problem, where the expectation of the rewards for each arm decays with respect to the number of pulls. The authors present natural settings motivating this variation, such as displaying ads to a user. The first authors cover the non-parametric case, and describe an algorithm using sliding window average only (no UCB). They prove a regret in O(T^2/3 log T). The author then cover the parametric case, where the expected rewards are parametrized by a known finite set of models. When the rewards decay to zero, they give a "plug-in" algorithm that obtain o(1) regret under certain assumptions. When the rewards decay to positive constant, they give a "UCB" algorithm obtaining O(log T) regret under certain assumptions. The author finally show results on empirical experiments. The core article is well written and the concepts are easy to follow. Since this setting is novel the article does not often use high level known results in their proofs, which make the theoretical analysis fairly technical. An important limitation is the necessary assumptions in the parametric case (Assumptions 4.2 and 4.3). First, these assumptions are hard to verify even for simple parametric models. (In the example analysed by the author, one has to wait for n~10^800 to observe bal(n) < n.) Second, the authors do not comment the generality of these assumptions. It seems that they are necessary for the current theoretical approach only, and modifying the proof would requires different assumptions.